# A Mathematical Model of Pressure Ulcer Formation to Facilitate Prevention and Management

**DOI:** 10.3390/mps7040062

**Published:** 2024-08-13

**Authors:** Ioannis G. Violaris, Konstantinos Kalafatakis, Nikolaos Giannakeas, Alexandros T. Tzallas, Markos Tsipouras

**Affiliations:** 1Department of Electrical and Computer Engineering, University of Western Macedonia, 50131 Kozani, Greece; john.violaris@yahoo.com (I.G.V.); mtsipouras@uowm.gr (M.T.); 2Faculty of Medicine and Dentistry (Malta Campus), Queen Mary University of London, VCT 2520 Victoria, Malta; 3Human Computer Interaction Laboratory (HCILab), Department of Informatics and Telecommunications, School of Informatics and Telecommunications, University of Ioannina, 47100 Arta, Greece; giannakeas@uoi.gr (N.G.); tzallas@uoi.gr (A.T.T.)

**Keywords:** pressure ulcers, mathematical model, differential geometry, system of ordinary differential equations

## Abstract

Pressure ulcers are a frequent issue involving localized damage to the skin and underlying tissues, commonly arising from prolonged hospitalization and immobilization. This paper introduces a mathematical model designed to elucidate the mechanics behind pressure ulcer formation, aiming to predict its occurrence and assist in its prevention. Utilizing differential geometry and elasticity theory, the model represents human skin and simulates its deformation under pressure. Additionally, a system of ordinary differential equations is employed to predict the outcomes of these deformations, estimating the cellular death rate in skin tissues and underlying layers. The model also incorporates changes in blood flow resulting from alterations in skin geometry. This comprehensive approach provides new insights into the optimal bed surfaces required to prevent pressure ulcers and offers a general predictive method to aid healthcare personnel in making informed decisions for at-risk patients. Compared to existing models in the literature, our model delivers a more thorough prediction method that aligns well with current data. It can forecast the time required for an immobilized individual to develop an ulcer in various body parts, considering different initial health conditions and treatment strategies.

## 1. Introduction

Pressure ulcers are sites of localized but permanent tissue damage, involving the skin and underlying histological layers, usually occurring over bony structures (for instance the sacrum, coccyx, heels, and hips) as a result of long-term pressure, such as being hospitalized for a prolonged period of time or being immobile and constantly lying on a bed with minimal positional changes. The pathophysiology of pressure ulcers involves obstructed blood flow to the soft tissues (either independent or in the context of an underlying pathology such as atherosclerosis), protein–calorie malnutrition, microclimate (such as skin wetness), and conditions that reduce the sensation of the skin (such as paralysis or neuropathy). The restoration of pressure ulcers may sometimes be really challenging, depending on the age of the individual, concurrent medical conditions, or smoking and medications.

In this work, we introduce a mathematical framework and a subsequent model with which to describe the process of pressure ulcer formation. We begin by introducing a formalism based on differential geometry to describe the surface of the human skin. Although tedious, we believe that such an approach can offer insights into pressure ulcer formation that would be impossible through the lens of a more simplistic description. Through the results of our model, it can be shown that any bed surface with a surface isometric to the plane would lead to skin deformations resulting in hypoxia. Another contribution of our model is the offering of a predictive method for pressure ulcer occurrence, which could aid medical professionals in taking timely preventive measures. This implies a different approach to mattresses specially designed to combat or reduce pressure ulcer formation towards the kinds that have a metric as close to the one of the human body as possible. Finally, we adapt our framework into a usable computer code to aid in the creation of future technologies aimed at solving this problem.

## 2. Materials and Methods

### 2.1. Mathematical Formalism

We assume that the human skin is a two-dimensional membrane. We will approach this as a surface as defined here:

**Definition** **1.** 
*We define a regular surface as a subset $S\subset R^{3}$ such that*

*$\forall p_{1},p_{2}\in S$: there exists an open set V⊂S such that $p_{1},p_{2}\in V.$*

*∃X:U→U∩S, $U\subset R^{2}$, X is surjective.*

*X is $C^{\infty }$ on U.*

*X is a homeomorphism.*

*∀q∈U, $dX_{q}:R^{2}\to R^{3}$ is injective.*



To estimate how the patch of skin is deformed under the weight of the body, we use the approach of energy minimization. We assume that the skin is a surface without thickness that would bend under the weight of the body in all directions with the bending in the perpendicular direction being bounded by the initial position and the distance of that position to the bone or other rigid structure of the body. Hence,
$$\vec{z}=\vec{n}=\frac{u\times v}{\left|u\times v\right|},\lambda \left|\vec{z}\right|\in \left[\lambda_{1},\lambda_{2}\right],$$
for some $\lambda ,\lambda_{1},\lambda_{2}\in R,$ uniquely defined by the position of the body being under consideration. In other words, the length that the skin may bend is constrained by the underlying structures of the body in each location.

In our case, the suggested energy functional has the form
(1)$$W\left(u,v,z\right)=Y\int_{V}\frac{1}{a}\sigma f\tilde{X}dV,$$

Here,
$$f=\left(\begin{array}{ccc} X_{uuu} & \left(X_{uuv}+X_{uvv}\right) & 0\\ \left(X_{uuv}+X_{uvv}\right) & X_{vvv} & 0\\ 0 & 0 & 0 \end{array}\right)$$
are the components of strain and
$$\sigma =\left(\begin{array}{ccc} -\frac{\nu }{Y} & -\frac{\nu }{Y} & \frac{1}{Y} \end{array}\right),$$
representing the material’s resistance to elongation towards each direction, while
$$\tilde{X}=\left(\begin{array}{c} X_{u}\\ X_{v}\\ X_{z} \end{array}\right),$$
defines the surface on which this force acts on (i.e., the tangent vector to *X*). Our choice of a 2-dimensional surface makes the terms $f_{13},f_{23},f_{33},f_{31},f_{32}$ equal to zero, simplifying the problem (for details about how this function was found, see The Theory of Elasticity [1] and Variational Methods in Elasticity and Plasticity [2] as well as the Appendix A). The indices here represent partial derivatives with respect to *u*, *v*, and z. Furthermore, ν is Poisson’s ratio for skin and *Y* is the material’s modulus of elasticity in tension. We chose to represent human skin as an isotropic material with a constant Poisson’s ratio and Young’s modulus to simplify the problem. Though not entirely realistic, it does not affect the result significantly enough when the problem is approached through this method, since the anisotropies would have to be extensive to affect the result in the scale and forces that are under consideration. The variable *a* is defined as the area of the surface $S_{1}$:$$S_{1}=\left\{\left(x,y,z\right)\in R^{3}:\left(x,y,z\right)\in P\cap X\left(u,v\right)\right\},$$
where *P* is the plane defined by the surface of the bed. To explain this intuition better, what happens is that at the beginning, the surface of the human body has a smaller area of intersection with the flat plane of the bed since we took into consideration that it is not flat. As it receives the force of its own weight and deforms, its area gets larger and has a tendency to minimize the received force by area as a direct result of the principle of the least action. Conversely, we consider it a safe assumption that beds made by most materials change their shapes from a flat plane to a cylinder, keeping their metric unchanged. Though trivial as a result, it is worth noting that since the area per arc length of a curve parameterized per arc length is equal to
$$\int_{A}dA=\int_{A}\sqrt{X_{u}^{2}X_{v}^{2}-{\left(X_{u}X_{v}\right)}^{2}}dudv\leq \int_{A}|X_{u}\left|\right|X_{v}|dudv,$$
the tendency would be toward the deformation maximizing the contact area, i.e., toward a surface isometric to a plane. Behind this, there is an important assumption of our model. The assumption is that the bed’s surface does not deform to accommodate the body and that it merely gets pushed down, taking the shape of a cylinder and leaving its metric unchanged. Equation (Equation 1) constitutes an assumption that we made based on the cited bibliography. It provides the work of deformation. As happens with similar problems, the area which minimizes the work is the area where we are trying to find the shape that the skin will take after receiving pressure. In doing so, we can then estimate how that will impact blood microcirculation and hence the survivability of the afflicted tissues. In Figure 1, we portray the example of a surface and how the stresses act on it to deform it.

It follows that
(2)$$W\left(u,v,z\right)=\left(\lambda_{2}-\lambda_{1}\right)Y\int_{A}\frac{1}{a}\sigma f\tilde{X}dA.$$

The term dA is by definition equal to
$$dA=\sqrt{EG-F^{2}}dudv,$$
where E,F, and *G* are the terms of the first fundamental form characterizing the metric of the surface in question. Finally, from this functional, we obtain the Euler–Lagrange equation of the form
(3)$$\sum_{i=0}^{2}\frac{\partial }{\partial x_{i}}\frac{\partial W}{\partial X_{i}}=0,$$
where $x_{0}=u,x_{1}=v,x_{2}=z$, and $X_{0}=X_{u},X_{1}=X_{v},X_{2}=X_{z}$ (proof in the Appendix B). For the definitions of e,g,f,E,G,F, see Appendix C [3]. Now, given the deformation of every point of the skin’s surface, we will estimate the effect that this has on blood circulation and finally use a system of ordinary differential equations (ODEs) to predict the rate of cellular death at each point. A particular solution to this partial differential equation would take the following form (graphically), as seen in Figure 2 (see the Appendix D).

Another effect of deformation is the change in angles that happens as a result of a change in the metric of the skin. This change would force capillaries to become warped. We can estimate this using the Gauss−Bonnet theorem and considering geodesic triangles inscribed within the circle on which the pressure acts. The Gauss−Bonnet theorem being a theorem about geodesic triangles inscribed on any surface (geodesic here implying following the least traversable path between two points) allows us to find out the new angles of the new shape that is created after pressure is applied. This is important since it allows us to see how much capillaries have been warped and hence calculate what effect that might have on the blood flow. An example of this appears below in Figure 3.

Using the fact that by Gauss−Bonnet, we have
$$\int \int_{T}kdA=a_{1}+a_{2}+a_{3}-\pi ,$$
we can estimate the difference in the angle to the initial triangle and find the angle between the old shape of the capillary and the new one. This would allow us to use fluid mechanics to find the changes in blood flow (the variable *k* denotes the Gaussian curvature).

Trying to find the change in blood flow in the capillaries from first principles due to their deformation using the laws of fluid mechanics is something beyond the scope of this paper. Thus, for our purposes, we will use Topakoglu’s [4] approximation (see the Appendix E). Hence, as long as we know the stresses on each part of the body’s surface, we can estimate the new form that would be taken by the body to accommodate its weight on the bed. This result is used to estimate the occurrence of hypoxia in tissues of the skin. To do that, we introduce a new diffusion model based on the works of Vivek et al. [5] and Wang [6].

### 2.2. Cellular Death Model

Now, since we estimated on which patches of skin there is a reduction in the partial pressure of oxygen as well as its extent, we will use the following system of ordinary differential equations (ODEs) to evaluate the damage being conducted in real time as well as how much time is left before the patient’s body is in danger of pressure ulcer formation:(4)$$\dot{N}\left(t\right)=\left(r_{0}-k_{N_{1}}I\left(t\right)\right)\left(1-\frac{N(t)}{K_{0}}\right)H\left(p\right)N\left(t\right),$$
(5)$$\dot{I}\left(t\right)=k_{N_{2}}I\left(t\right)\left(1-\frac{I(t)}{K_{1}}\right)+k_{D_{0}}D\left(t\right),$$
(6)$$\dot{D}\left(t\right)=\left(1-\frac{N(t)}{K_{0}}\right)k_{D_{1}}-k_{D_{2}}D\left(t\right).$$
where *N* denotes the normalized cell population of the patch of skin under study, N∈[0,1]; $r_{0}$ is the reproductive rate of cells (constant); H(p) is a Hill function dependent on the stress received by each point; *p* is the oxygen partial pressure resulting from that; and I(t) is the density of neutrophils, $K_{0},K_{1},k_{N_{1}},k_{N_{2}},k_{D_{0}},k_{D_{1}},k_{D_{2}}$ constants. The function D(t) symbolizes the density of “damage associated molecular patterns” or “Damps”, which affects the immune system reaction. In other words, what is assumed is happening is that as cells die, they produce “debrie”, which in turn mobilizes the immune system (the same assumption holds in Vivek et al. [5]). We decided to take into account the immune response based on considerations made in Vivek et al. [5], though we took a simpler approach to solving the problem.

We also decided to solve for the normalized functions (see the Appendix F), N(t),I(t), D(t), because the quantity of interest is the percentage of initial cells still remaining alive, which would show a continuous path toward pressure ulcer formation. At the end of this subchapter appears a table including all the variables used in our equations, their meaning, as well as the used values in the simulations.

The numbers used have either been taken from V. D. Sree et al. [5] or estimated via trial and error to produce physically relevant simulations. The variable $K_{1}$ is a bit higher than it should be due to it needing to appear separate from zero in the graphs. It is only an indicative value and does not affect the physical relevance of the graphs. In Figure 4 we have included all the parameters used in the model and their values.

### 2.3. Surfaces of Interest

A question that arises based on our approach is what kinds of surfaces can be used to approximate the shape of the human body when no external forces are acting on it. We find that it is a realistic and simplifying approximation to approach the human body as the union of a set of known shapes, giving more attention to areas where pressure ulcers usually occur like the back, the pelvis, or the back of the head. The body is symmetrical and hence the part starting from the pelvis and reaching the shoulders can be seen as a surface of revolution similar to the upper half of a catenoid described by
X(u,v)=(acosh(v/c)cos(u),bcosh(v/c)sin(u),v),
with 0≤u≤π and 0≤v≤0.34 m.

Examples of surfaces approximating the shape of a human back can be seen below in Figure 5.

For the purposes of this paper, we will only concern ourselves with the part of the human back that most commonly suffers from pressure ulcers. A similar study can, in theory, be made for any other part of the body.

## 3. Results

Having developed a model to handle a large set of situations that might lead to the formation of pressure ulcers (Section 2), we will apply it on areas where we typically have such a development. Starting by modeling each part of the body based on some known parametric curve, we have that most of the parts in question can be thought of as parts of a catenoid, as is shown in Figure 6.

To produce the results, we observed the following procedure:Estimate the area of the surface of an average human body touching the plane of the bed.Use the theory we established in the Materials and Methods to calculate how the skin in these areas would be deformed after receiving the pressure from its own weight and finding the new post-deformation area and Gaussian curvature.Calculate the capillary density of the new deformed surface as well as the reduction in blood flow due to the new curvature.Use our proposed model of ODEs to predict the effect of that deformation on the cells of the epidermis over time over the areas in question.

This chart (Figure 7) portrays how the process allows us to produce the required results.

The area around the second vertebra on the back is sensitive to pressure ulcer formation. We decided to focus on an area that is $\frac{1}{16}$th of the whole back of an average male. The shape used is shown below in Figure 8.

The area in question is assumed to be equal to 0.0121318 m^2^ on average when no forces are acting on it (i.e., no deformation has yet occurred). Based on A. Delalleau et al. [8], we can assume that the average Young’s modulus for human skin is 0.5 MPa, while its Poisson’s ratio can be assumed to be 0.495 (i.e., close to a perfectly incompressible elastic body). Based on this, we can use the general Hook’s law [1] to estimate the elongation of the boundaries of the surface in question. Doing the calculations, we have that
$$\epsilon_{uu}=\epsilon_{vv}=\frac{1}{Y}\left(-\nu \sigma_{zz}\right).$$
which results in an extension of 0.95 mm of the boundaries of the surface in question on its minimum diameter (i.e., $\frac{1}{2}\times 0.475$ mm in each direction).Remark: The force acting perpendicular to the surface of the skin causes the skin to contract in the *z* direction, resulting in an extension in the *u* and *v* directions. Note that we also assumed that the whole weight of the pelvis (14% of the body’s weight) falls on that surface, resulting in 10.5 kg for an average adult male (the average weight is taken to be 75 kg).

We can also hypothesize that the skin indentation in the *z* direction is equal to 0.048 mm assuming a skin depth of 2.75 mm (the average of the known values from [9]). Based on this, we have initial and boundary conditions for the new curve X(u,v). Its length *L* must be 0.95 mm longer than the initial curve under consideration and its depth 0.048 mm greater at its central point again than the initial curve (for an average male adult human). We will further assume, as this seems to be the case, that although the metric of the skin will change as predicted by our model, the boundaries will be the elongated boundaries of the initial catenoid. This is based on the simple observation that although the skin is deformable, its boundaries seem to always keep roughly the same shape as before excluding places of the body where the bones are also deformable (like the nose or ears), places that are not studied here. Our assumptions lead to the following boundary conditions for the problem:$$X\left(0,v\right)=\left(A\cosh\frac{v}{c},0,\left(v-d\right)\right),$$
$$X\left(\pi ,v\right)=\left(-A\cosh\frac{v}{c},0,-\left(v-d\right)\right).$$
$$X\left(u,0\right)=\left(A\cos u,B\sin u,-d\right)$$
$$X\left(u,D\right)=\left(A\cos u\cosh\frac{D}{c},B\sin u\cosh\frac{D}{c},D-d\right)$$

Under the pressure of its own weight, the area of an average human back around the second vertebra would then take the following shape:$$X\left(u,v\right)=\left(A\cosh\frac{v}{c}\cos\frac{u}{c},B\cosh\frac{v}{c}\sin\frac{u}{c},v-d\right)$$

It turns out that the chosen shape of the back solves the biharmonic equation. To show this, it is only required to notice that if we perform a change in parameters such that
u→cu
the shape satisfies a solution of the biharmonic, as stated in Polyanin et al. [10]. The new shape one acquires after this is shown below in Figure 9. It appears almost identical to the previous one, but comparing the two in the same image demonstrates the existing differences between them.

The parameters A,B,d are
$$A=a+a\epsilon_{uu}=0.0572\;m,$$
$$B=b-b\epsilon_{vv}=0.02744\;m,$$
$$D=D^{\prime }+D^{\prime }\epsilon_{zz}=0.092\;m,$$
d=0.00048m,
and *c* is the same as in the initial shape. This new shape has a different area, which is estimated as
$$A^{\prime }=0.0130116{\;m}^{2}$$

For comparison, we present the two surfaces together in Figure 10.

This change implies a reduction in the density of the capillaries ρ by
$$\frac{\rho }{\rho^{\prime }}=\frac{A}{A^{\prime }}=11.9\%.$$

Now, we also have to take into account the change in the Gaussian curvature
$$k=\frac{eg-f^{2}}{EG-F^{2}},$$
due to the change in the parameters of the curve.

Topakoglu’s approximation implies that the reduction in flow would be dependent on the change in curvature at each point. Since the curvature is already positive and non-zero from the start, we have to compare both the initial and after-deformation curvatures to find the degree of flow reduction. To explain that, before deformation, we have a curvature $k_{1}$, which becomes $k_{2}$ after the deformation. That would imply that in relation to a straight capillary before the deformation, we would have
$$\frac{Q_{cT_{1}}}{Q_{sT}}=1-\frac{1}{48}{\left(\delta_{1}\right)}^{2}\left(\frac{1.541}{67.2}n^{2}+1.1n-1\right),$$
while after the deformation, that would change to
$$\frac{Q_{cT_{2}}}{Q_{sT}}=1-\frac{1}{48}{\left(\delta_{2}\right)}^{2}\left(\frac{1.541}{67.2}n^{2}+1.1n-1\right),$$
implying that
$$\frac{Q_{cT_{1}}}{Q_{cT_{2}}}=\frac{1-\frac{1}{48}{\left(\delta_{2}\right)}^{2}\left(\frac{1.541}{67.2}n^{2}+1.1n-1\right)}{1-\frac{1}{48}{\left(\delta_{1}\right)}^{2}\left(\frac{1.541}{67.2}n^{2}+1.1n-1\right)},$$

Doing the necessary calculations, we find that
$$k_{1}=\frac{\cosh^{2}\left(0.267v\right)\left(0.00008\sin^{2}\left(2u\right)\sinh^{2}\left(0.267v\right)+0.071\cos\left(2u\right)+0.128\right)}{(0.464\cos(2u)-0.961)\cosh(0.533v)+0.535\cos(2u)-1767.57}$$
i.e., that is the formula that predicts the Gaussian curvature at every point (u,v) when no forces are acting on the body. Similarly, doing the same calculations for $k_{2}$, i.e., the Gaussian curvature of the skin after the deformation has occurred, we find
$$k_{2}=\frac{\cosh^{2}\left(0.267v\right)\left(-0.000056\sin^{2}\left(0.533u\right)\sinh^{2}\left(0.267v\right)-0.0445\cos\left(0.533u\right)-0.071\right)}{-0.6258\cos(0.533u)+\cosh(0.533v)+13,975.9}$$

Based on Timothy W. Secomb’s [11] work, we may assume that the Reynold’s number is approximately 7 × 10^−3^ for an average microvessel with a diameter of 19.5 μm. Both $k_{1}$ and $k_{2}$ vary across their respective curves as is expected. It is reasonable to expect that the reduction in flow would be dependent upon their maximum absolute values. Plotting $|k_{1}|$ and $|k_{2}|$ (Figure 11) shows that there exists no unique value of *v* that satisfies the definition of a global maximum.

Thus, we would have to estimate the supremum of $k_{1}\left[K_{1}\right]$ and $k_{2}\left[K_{2}\right]$, where $K_{1}=\left\{p\in R^{2}:p=k_{1}\left(u,v\right),u\in \left(u_{1},u_{2}\right),v\in \left(v_{1},v_{2}\right)\right\}$ is the set of the image of $k_{1}$, and with a similar definition, $K_{2}$ is the set of the image of $k_{2}$ as
$$\sup\left(K_{1}\right)=1.1\times 10^{-3},$$
and
$$\sup\left(K_{2}\right)=8.7\times 10^{-6}$$
respectively.

Using Topakoglu’s approximation and conducting the calculation yields a result of approximately one, i.e., there is no significant reduction in the flow due to the bending of the tubes. Hence, we may proceed with estimating the effect that the change in area yields to the distribution of oxygen amongst the tissues ignoring any changes in flow.

As can be seen in Figure 12, a healthy individual without any underlying health problems would, under total immobilization on a standard bed, start developing visible tissue damage after only about 10 days. This damage, if left untreated, would result in a total destruction of the skin’s tissue in around 20 days from total immobilization. Assuming other pathological conditions that reduce capillary blood flow such as smoking [12] or diabetes melitus [13] and starting from an initial $O_{2}$ pressure that is lower than 40 mm Hg would induce damage quicker, as shown in Figure 13.

Here, assuming a starting point with just 15% lower $O_{2}$ partial pressure than the ideal conditions, we show the onset of pressure ulcer formation in only 3 days, something that agrees with some extreme cases in empirical data [14]. If we assume treatment where the position of the patient is changed every 2 h, as is the common practice, that would change the result as follows in Figure 14 and Figure 15.

As can be seen in previous figures, changing the position of the patient can dramatically increase the time that is needed for a pressure ulcer to form but not stop it completely. Given that, based on Schoonhoven et al. [14], only about 6% of patients develop an ulcer in the first week, we can see that this result also agrees with the data as only people with underlying circulatory problems are liable to have an ulcer quickly, as shown by the model. Also, as shown in Bours et al. [15], the mean days from admission to the onset of pressure ulcers are 16 and 19 days for university and public hospitals, respectively. These values are well in agreement with our model since no patient is left without any treatment, but a lot of times treatment is not as ideal as what we simulate.

The different times that it takes individuals to develop a pressure ulcer in similar conditions can be attributed, at least based on our model, to individual differences in weight, bone density, individual differences in the skin’s Young’s modulus, underlying circulatory problems, and the initial skin condition. If we start assuming small wounds on the body, pressure ulcer development becomes faster. A key conclusion that can be deduced from the results is that given enough time and a bed surface isometric to the plane as the one we assumed in the model’s creation, everyone will at some point develop an ulcer, though for some people the time might be much longer than their actual immobilization.

## 4. Discussion

Pressure ulcers constitute a problem resulting from long-term immobilization in healthcare units. Their consequences are dire since they result in longer hospitalization times and an increase in hospitalization and treatment costs while putting a lot of pressure on healthcare personnel. Various studies have been conducted to estimate their economic cost in different healthcare systems. Padula et al. [16] estimate that the cost for the US national healthcare system exceeds USD 26.8 billion yearly, while a metanalysis and systematic review [17] shows that at least one in ten hospitalized adults suffer from pressure ulcers during their hospitalization.

Their etiology has not been sufficiently modeled, and although various methods of combating them exist, no method seems to completely solve the problem. The bibliography usually focuses on particular aspects of pressure ulcer formation [5,18,19] like the interplay between ulcer and inflammation, risk prediction, or the appearance of ulcers in a particular class of patient. Many aspects of the problem have been analyzed with mathematical modeling [20], but the nature of the surface on which the human body rests on is almost never considered in a general way. With the development of this model, we attempt to shed light on their etiology in order to help the research community find ways to more effectively combat their appearance. We attempted to create a model which is general enough to be capable of being implemented on different parts of the body in contact with different surfaces. Our model is also capable of predicting the time required for pressure ulcer formation in wildly different conditions while taking into account individual parameters relating to patients with other underlying health conditions.

Our model manages to predict the formation of pressure ulcers and give an explanation of the underlying mechanisms responsible for this condition. Its downside is that it is complex and hard to implement because the individual Young’s moduli and Poisson’s ratio for every possible patient are not known. The calculations involved are also extremely complex, though the time that is needed can be changed by a modern computer in seconds. Yet human judgment is required, and automating the process would need more research to be conducted. High-quality data giving a description of pressure ulcer formation and its relation to time is also absent from the bibliography, something which denied us the data necessary to properly compare our results to reality. Hence, we believe that more data of that sort should be collected before we can refine our method to more reliably predict pressure ulcer formation. Another weakness of our model is the fact that we have not considered specially designed bed surfaces, which would distribute forces more evenly. We made this choice in order to show the importance of the metric of the bed surface when considering what kinds of deformable materials should be used.

In addition, we used a simplified approach to modeling the human skin, considering it an isotropic elastic material. In state-of-the-art models, the skin is considered a hyperelastic anisotropic material [21]. Yet, if the underlying mathematics of this approach are studied, it becomes obvious that there are no significant differences in the predictions between these models and our own. We had to make a compromise between simplicity and ease of computation and realism to make our model easier to use and implement. Our model would fail, for example, if the acting pressure on the skin is assumed in more complex conditions like a body on a pillow lying on a bed. Also, we did not take into account microclimate factors and the interplay between the skin and underlying bony structures. Yet, as a comparison between the results and data shows, we still do predict the proper onset for pressure ulcer occurrence.

Artificial intelligence applications in mathematical modeling could improve our model considerably. For example, a method like the one proposed in Li et al. [22] could be used to create a model for skin deformation predictions, which does not suffer from the limitations that our model does. One issue with such approaches though would be the lack of insight into the mechanics, and hence the value of the model to someone wishing to create a mattress which delays ulcer appearance would lessen. On the other hand, its value as a predictive tool would be greater, so it could be used to evaluate such mechanisms in real life and lead to more efficient approaches. An additional potential application of artificial intelligence models to enhance our research is demonstrated by a study conducted by Yu et al. [23]. While their method addresses information cascades, pressure ulcers can similarly be conceptualized as a cascade involving sequential failures of tissue and the immune system due to prolonged pressure and other underlying health conditions. Therefore, employing such models could significantly advance our research and would represent a suitable direction for continuing investigations in this area or related topics.

The novelty of our approach lies in the fact that we considered the interplay between human skin and a mattress, something that has not been performed yet. Usually, either one or the other conditions are modeled in isolation. This interplay having been modeled could be useful in guiding engineers in their research to create mattresses which can reduce the probability of occurrence or even eradicate pressure ulcer occurrence altogether.

An adaptation to make predictions about different mattress materials is possible, and future works may tackle that problem as well. Our model can be useful to the engineer trying to create a bed or wheelchair which tries to minimize the probability of pressure ulcer formation. It shows how the interplay between bed and skin deformation results in a reduction in blood circulation, which finally results in cellular death. It can also be used as a predictive tool to help medical personnel take action to reduce the probability or average time of occurrence of pressure ulcers.

## Figures and Tables

**Figure 1 mps-07-00062-f001:**
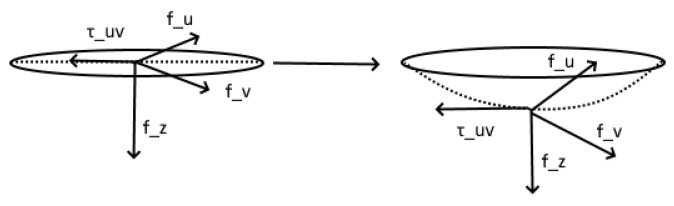
The surface under question along with the stress acting in all three directions.

**Figure 2 mps-07-00062-f002:**
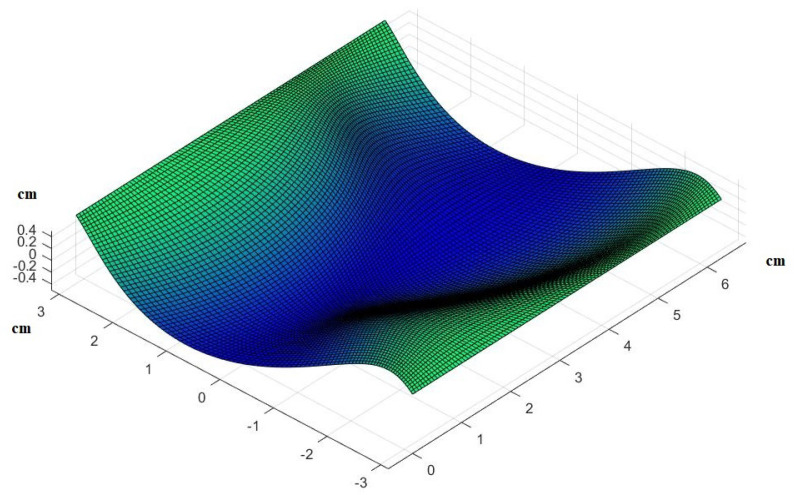
The form that human skin takes under pressure. Here we present an example of a particular solution. Assuming that something like, for example, a human finger pushes the skin and creates an indentation, we show how our model predicts it would look like.

**Figure 3 mps-07-00062-f003:**
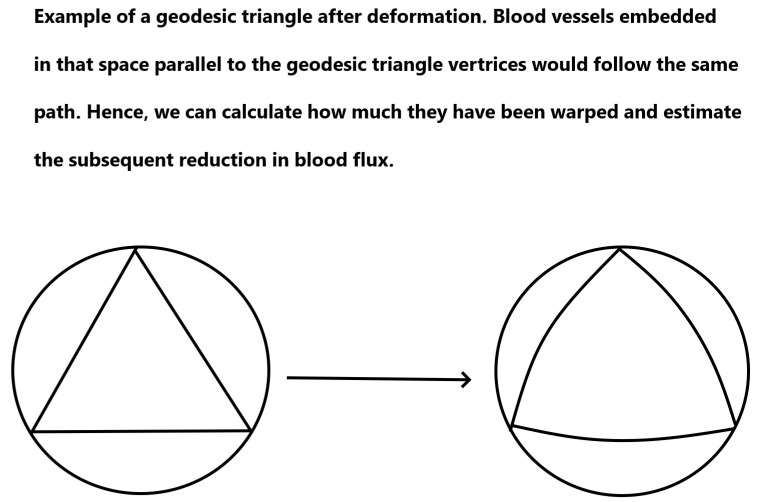
An example of geodesic triangles and how their vertices would curve under a change in the curvature of the surface they are inscribed upon.

**Figure 4 mps-07-00062-f004:**
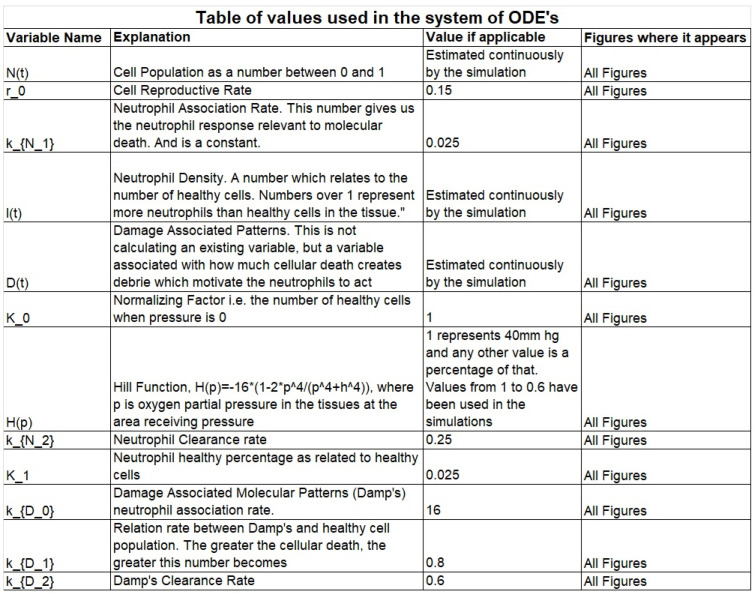
Explanations for the variables used in the ODE model are given as well as the values used.

**Figure 5 mps-07-00062-f005:**
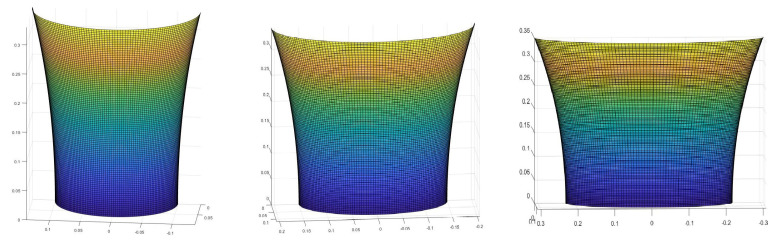
Examples of the surface of a human back based on measurements from the literature [7]. The axes are all in meters.

**Figure 6 mps-07-00062-f006:**
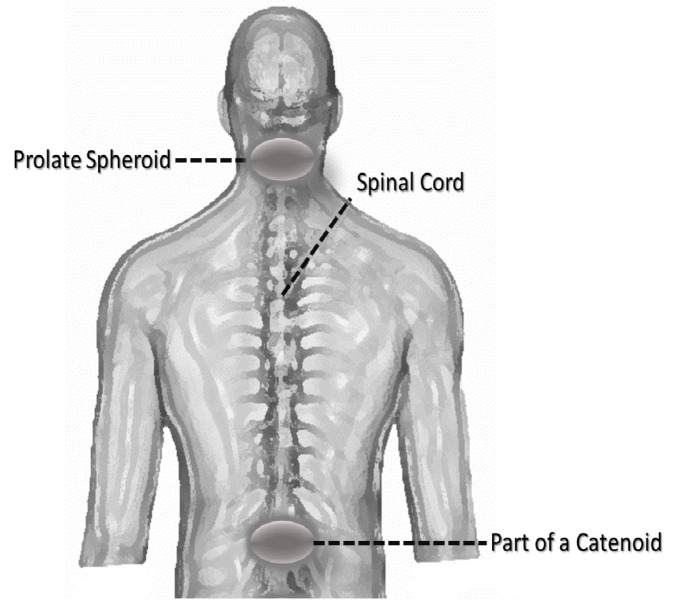
A figure of the human body along with areas where pressure ulcers usually appear and their close mathematical equivalents.

**Figure 7 mps-07-00062-f007:**
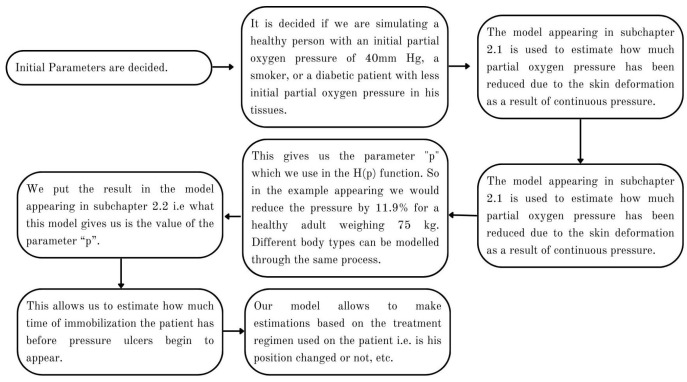
In this chart, the process through which the model is used is explained.

**Figure 8 mps-07-00062-f008:**
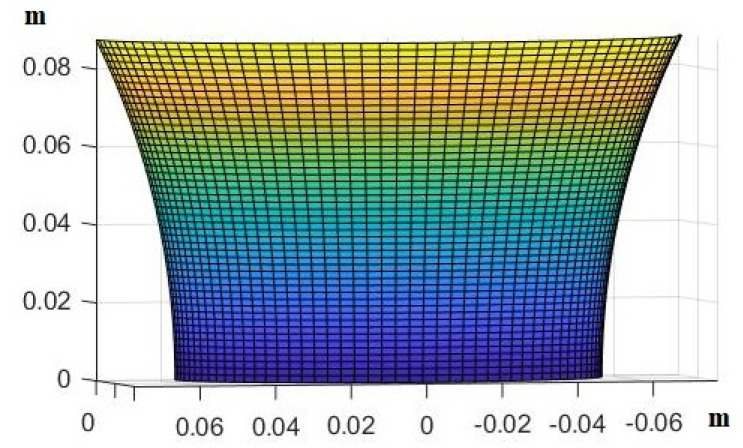
The shape of the assumed human back that we used in the modeling process.

**Figure 9 mps-07-00062-f009:**
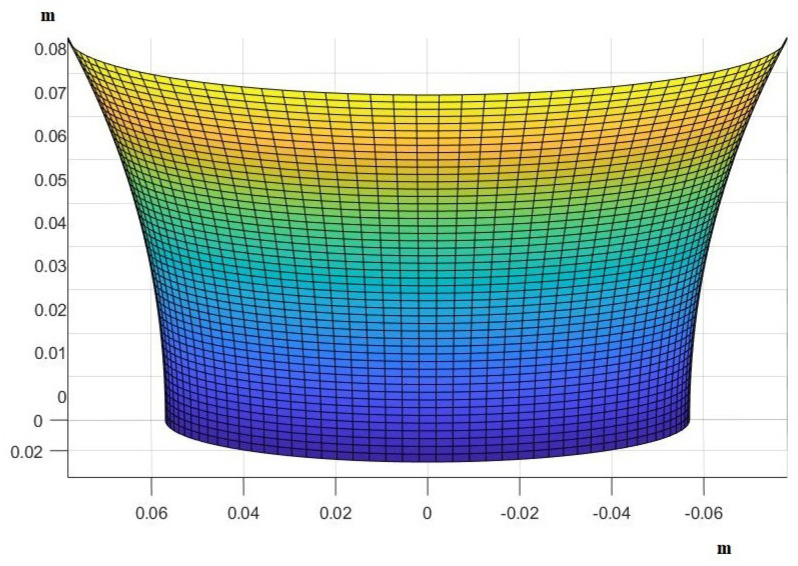
The shape that the back has taken after pressure is acting on it. Here, the difference is hard to see. This will become evident in the next figure when the two are put together for comparison.

**Figure 10 mps-07-00062-f010:**
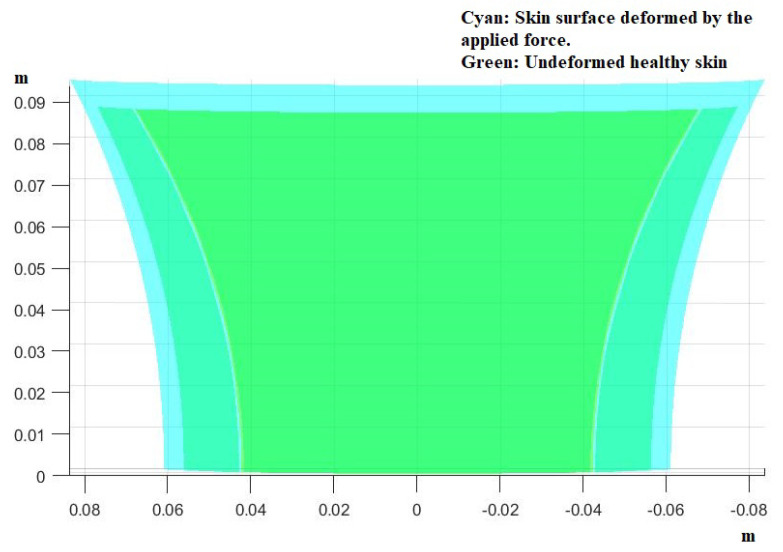
The two previous figures appearing together, one overlapping the other. It is evident that the area increased. This would not be so easy to see in real life due to the fact that the object in question is 3−dimensional and not 2−dimensional as it appears here.

**Figure 11 mps-07-00062-f011:**
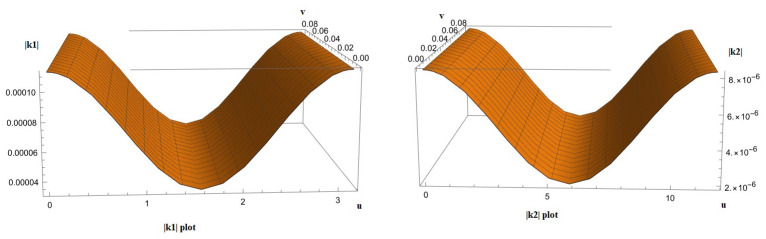
The plots of the Gaussian curvature of the surface of the human back at each point appear here.

**Figure 12 mps-07-00062-f012:**
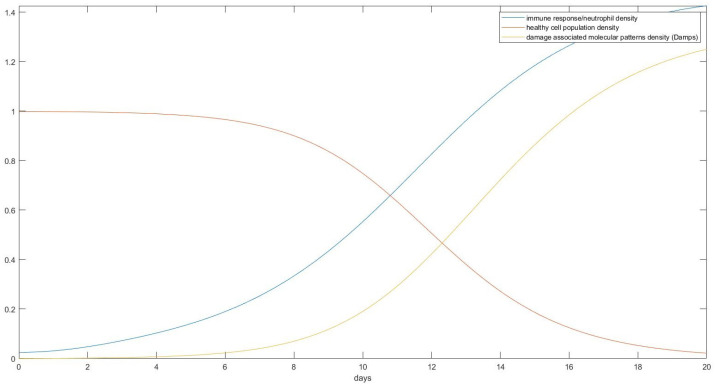
The prediction of the ODE model regarding cellular death as a function of time. This is the body of a perfectly healthy individual after immobilization without changing his position.

**Figure 13 mps-07-00062-f013:**
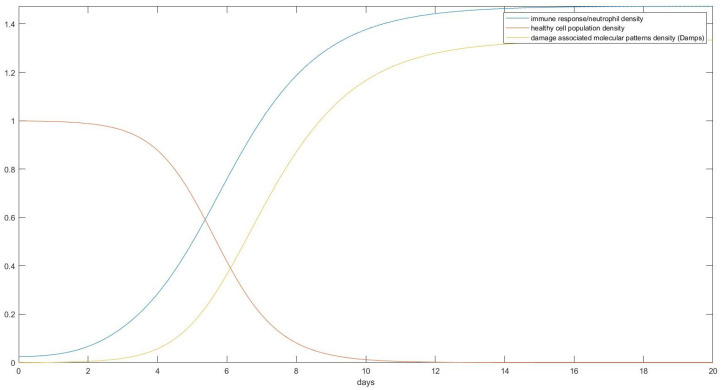
The prediction of our model on an individual whose initial partial oxygen pressure was reduced by 15% from the start due to other underlying health problems.

**Figure 14 mps-07-00062-f014:**
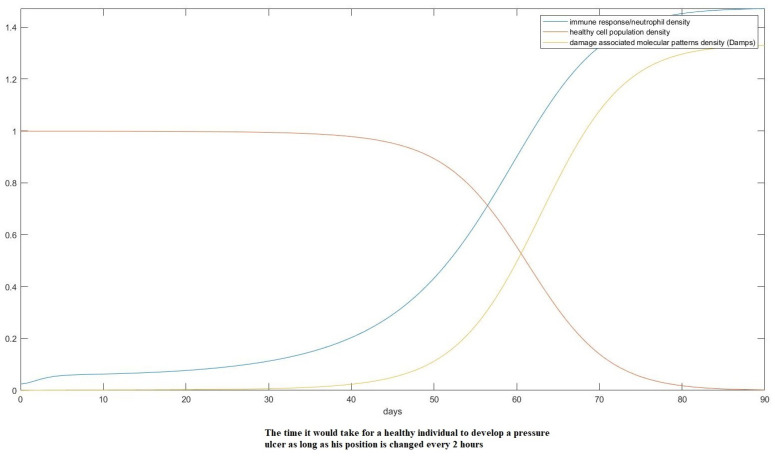
Model prediction of a healthy individual whose position is changed every 2 h. We used a step function to imitate the effect, changing the pressure to 1 every 2 h and then back to the predicted value when 2 h have elapsed.

**Figure 15 mps-07-00062-f015:**
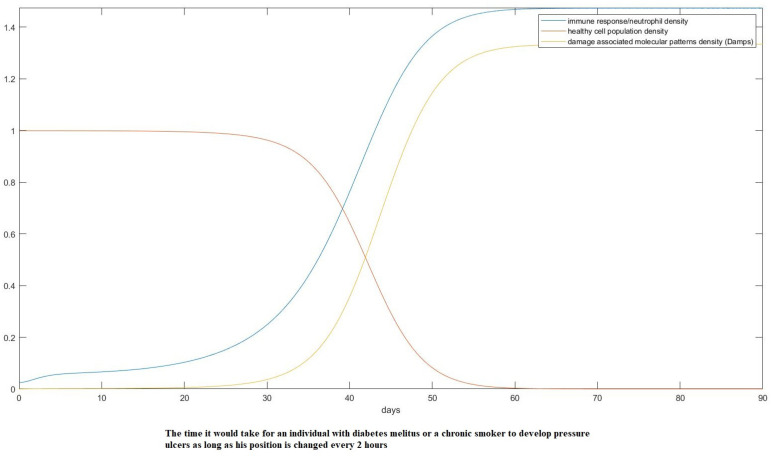
Model prediction of an unhealthy individual when pressure is alleviated every 2 h.

## Data Availability

All data are available in the manuscript and its appendices.

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
