# Peer review of "A Mathematical Model of Pressure Ulcer Formation to Facilitate Prevention and Management"

_mps, 2024, doi:10.3390/mps7040062_

Round 1
Reviewer 1 Report
Comments and Suggestions for Authors
Comments to the Author
The present work is original and interesting. However, the authors should answer the following comments to improve the quality of the article.
[1]. The English grammar and spell check needs to improve throughout the paper.
[2]. The presentation of the abstract is a little weak. Revise the abstract with a comprehensive overview of the study, clearly stating the research problem, methodology, key findings, and implications.
[3]. Check the governing equation once and cite a reference for it.
[4]. What is the significance of the current investigation?
[5]. Add novelty to the present work.
[6] Some physical explanations are missed.
[7] The presentation of all the equations is not uniform, and glitches are missed at some places.
Comments on the Quality of English Language
Can be improved.
Author Response
[1]. The English grammar and spell check needs to improve throughout the paper.
We made various minor corrections throughout the paper.
[2]. The presentation of the abstract is a little weak. Revise the abstract with a comprehensive overview of the study, clearly stating the research problem, methodology, key findings, and implications.
We have extensively revised the abstract, among others by adding the potential clinical utility of our model. This can be found in yellow on page 1 of the revised manuscript.
[3]. Check the governing equation once and cite a reference for it.
We have added further clarifications concerning equation 2.1. This can be found in yellow on page 3 of the revised manuscript.
[4]. What is the significance of the current investigation?
We have expanded the discussion section to provide more details on this. This can be found in yellow on pages 14-15 of the revised manuscript.
[5]. Add novelty to the present work.
We have expanded the discussion section to provide more details on this. This can be found in yellow on pages 14-15 of the revised manuscript.
[6] Some physical explanations are missed.
We have added a Table with all physical explanations. This can be found in yellow on page 6 of the revised manuscript.
[7] The presentation of all the equations is not uniform, and glitches are missed at some places.
We have corrected any minor issues with equation presentation, and now we believe all equations are presented uniformly.
Reviewer 2 Report
Comments and Suggestions for Authors
No terms used for the mathematical model are explained. This makes reading very cumbersome and an evaluation of the plausibility of the equations as well as the choices made for the parameters nearly impossible.
The authors state that some simplifications are not entirely realistic. They have to discuss the validity and limitations of their model in more details than indicated in the Discussion. How valid ist it to consider the skin as homogeneous layer and not consider the changes in mechanical properties of the different layers or some further influencing factors like shear forces or moisture accumulation?
A full validation of the model is probably difficult for in-vivo human skin under pressure. However, there exist some ex-vivo or in-vitro models of skin under mechanical load described in the literature. The authors should provide a minimum validation of their model prior to the acceptance of this paper for publication.
Author Response
No terms used for the mathematical model are explained. This makes reading very cumbersome and an evaluation of the plausibility of the equations as well as the choices made for the parameters nearly impossible.
We have added a Table (Figure 4, page 6 of the revised manuscript) with explanations about all terms and initial parameters used for the mathematical model.
The authors state that some simplifications are not entirely realistic. They have to discuss the validity and limitations of their model in more details than indicated in the Discussion. How valid is it to consider the skin as homogeneous layer and not consider the changes in mechanical properties of the different layers or some further influencing factors like shear forces or moisture accumulation?
Limitations of the model are mentioned in the Discussion section of the manuscript. This can be found in yellow on pages 14-15 of the revised manuscript.
A full validation of the model is probably difficult for in-vivo human skin under pressure. However, there exist some ex-vivo or in-vitro models of skin under mechanical load described in the literature. The authors should provide a minimum validation of their model prior to the acceptance of this paper for publication.
We cite 1 study (Bours et al [19]) which indicates that the mean days from admission to onset of pressure ulcers are 16 and 19 days in university and public hospitals respectively. These values are well in agreement with our model, since no patient is left without any treatment, but a lot of times treatment is not as ideal as what we simulate. (In yellow, on page 13 of the revised manuscript)
Reviewer 3 Report
Comments and Suggestions for Authors
The work spans across disciplines, integrating insights from electrical and computer engineering, medicine, and informatics. The authors utilize differential geometry and elasticity theory to model the deformation of human skin under pressure, offering a new perspective on the mechanics leading to pressure ulcers. This innovative modeling technique considers the geometric changes in the skin and their impact on blood flow, a critical factor in ulcer formation. By incorporating a system of ordinary differential equations, the model estimates the rate of cellular death in affected skin and underlying tissues, providing a comprehensive predictive framework. However, there are a few areas where improvements could be made for increased clarity and professionalism:
1. The citations provided lack a consistent style throughout the text. While specific sources are mentioned within the text, they are not formatted uniformly according to a recognized citation style (e.g., APA, MLA, Chicago), which can confuse readers and hinder their ability to locate referenced materials.
2. While abbreviations can streamline text, excessive use without prior definitions can be confusing. For instance, terms like "ODE's" for ordinary differential equations should be defined upon first use to ensure readability for a broad audience.
3. The text seems to blend sections together without clear paragraph breaks, which can affect readability. Each new concept or subtopic should ideally begin with a new paragraph for clarity. And the manuscript includes technical terms and equations without adequate explanation for a non-expert reader. Concepts such as Gaussian curvature (k) and the Gauss-Bonnet theorem should be briefly explained or accompanied by references for further reading.
4. Figures are mentioned (e.g., Figure 2, Figure 3, etc.) without providing descriptive captions within the text. This makes it difficult for readers to visualize the data or understand the relevance of the figures to the discussion.
5. In some instances, equations are described within the text without proper formatting. For example, Equation (4) and (5) related to cellular death models are presented in plain text rather than being formatted as distinct equations, which could hinder comprehension.
9. The article requires a visual structural diagram to clearly illustrate how the mathematical model corresponds to the mechanical mechanisms underlying the formation of pressure ulcers.
10. The authors ought to discuss the most recent advancements in deep learning techniques and their applications within mathematical modeling and prediction, for instance, the prediction and the numerical solution of mathematical models such as differential equations (PDEs), deep learning methodologies, have made significant strides in addressing these issues effectively (Yu, Dingguo, et al. "Information cascade prediction of complex networks based on physics-informed graph convolutional network." New Journal of Physics26.1 (2024): 013031.), complementing traditional approaches with their capacity for capturing complex structures and dynamics.
Comments on the Quality of English Language
none
Author Response
- The citations provided lack a consistent style throughout the text. While specific sources are mentioned within the text, they are not formatted uniformly according to a recognized citation style (e.g., APA, MLA, Chicago), which can confuse readers and hinder their ability to locate referenced materials.
We revised the citations to follow a uniform style.
- While abbreviations can streamline text, excessive use without prior definitions can be confusing. For instance, terms like "ODE's" for ordinary differential equations should be defined upon first use to ensure readability for a broad audience.
We corrected this accordingly. We defined all abbreviations upon first use.
- The text seems to blend sections together without clear paragraph breaks, which can affect readability. Each new concept or subtopic should ideally begin with a new paragraph for clarity. And the manuscript includes technical terms and equations without adequate explanation for a non-expert reader. Concepts such as Gaussian curvature (k) and the Gauss-Bonnet theorem should be briefly explained or accompanied by references for further reading.
We revised the manuscript accordingly to clearer show paragraph breaks. Moreover, we expanded the text to explain technical terms and equations like the Gauss-Bonnet theorem. (In yellow, on page 4 of the revised manuscript)
- Figures are mentioned (e.g., Figure 2, Figure 3, etc.) without providing descriptive captions within the text. This makes it difficult for readers to visualize the data or understand the relevance of the figures to the discussion.
Thank you for this note. We added reference to Figures and Tables within the text. (Indicated in yellow in the revised manuscript)
- In some instances, equations are described within the text without proper formatting. For example, Equation (4) and (5) related to cellular death models are presented in plain text rather than being formatted as distinct equations, which could hinder comprehension.
We revised the formatting appropriately.
- The article requires a visual structural diagram to clearly illustrate how the mathematical model corresponds to the mechanical mechanisms underlying the formation of pressure ulcers.
We have added this diagram. (Figure 7, page 8 of the revised manuscript)
- The authors ought to discuss the most recent advancements in deep learning techniques and their applications within mathematical modeling and prediction, for instance, the prediction and the numerical solution of mathematical models such as differential equations (PDEs), deep learning methodologies, have made significant strides in addressing these issues effectively (Yu, Dingguo, et al. "Information cascade prediction of complex networks based on physics-informed graph convolutional network." New Journal of Physics26.1 (2024): 013031.), complementing traditional approaches with their capacity for capturing complex structures and dynamics.
We have expanded the Discussion section to include these thoughts and citation. This can be found in yellow on pages 14-15 of the revised manuscript.
Round 2
Reviewer 2 Report
Comments and Suggestions for Authors
Thank you for considering the comments.
Author Response
Comment 1: Thank you for considering the comments.
Response: We would like to express our deepest gratitude to the Reviewer for helping us substantially improve the manuscript.
Reviewer 3 Report
Comments and Suggestions for Authors
I can accept this version, but please make the flowchart in the article clearer, without large blocks of text and with more specific steps. And other images are also not clear enough, please insert vector images.
Comments on the Quality of English Language
none
Author Response
Comment 1: I can accept this version, but please make the flowchart in the article clearer, without large blocks of text and with more specific steps. And other images are also not clear enough, please insert vector images.
Response: We would like to thank the Reviewer for his contribution to substantially improving our manuscript. We have modified the flowchart according to the suggestions (Figure 7 of the revised manuscript) and tried to improve the quality of all images of the manuscript.